# FAM83G Is a Novel Inducer of Apoptosis

**DOI:** 10.3390/molecules25122810

**Published:** 2020-06-18

**Authors:** Junichi Okada, Noriaki Sunaga, Eijiro Yamada, Tsugumichi Saito, Atsushi Ozawa, Yasuyo Nakajima, Kazuya Okada, Jeffrey E. Pessin, Shuichi Okada, Masanobu Yamada

**Affiliations:** 1Department of Medicine and Molecular Science, Gunma University Graduate School of Medicine, 3-39-15 Showa-machi, Maebashi 371-8511, Japan; m1720012@gunma-u.ac.jp (J.O.); eijiro.yamada@gunma-u.ac.jp (E.Y.); saitotsu@gunma-u.ac.jp (T.S.); ozawaa@gunma-u.ac.jp (A.O.); ynihei@gunma-u.ac.jp (Y.N.); myamada@gunma-u.ac.jp (M.Y.); 2Department of Respiratory Medicine, Gunma University Graduate School of Medicine, 3-39-15 Showa-machi, Maebashi 371-8511, Japan; nsunaga@gunma-u.ac.jp; 3Omagari Kosei Medical Center, 8-65 Omagaritori-machi, Daisen 014-0027, Japan; kazsgs0609@gmail.com; 4Department of Medicine and Molecular Pharmacology, Albert Einstein College of Medicine, Bronx, NY 10461, USA; jeffrey.pessin@einsteinmed.org

**Keywords:** family with sequence similarity 83 protein family G, heat shock protein 27, apoptosis, HSP27 phosphorylation, serine/threonine-protein kinase D1/protein kinase C mu

## Abstract

The family with sequence similarity 83 (FAM83) protein family G (FAM83G) possesses a predicted consensus phosphorylation motif for serine/threonine-protein kinase D1/protein kinase C mu (PKD1/PKCμ) at serine residue 356 (S356). In this study, overexpressed wild-type FAM83G coimmunoprecipitated with PKD1/PKCμ in Chinese hamster ovary (CHO) cells inhibited heat shock protein 27 (HSP27) phosphorylation at S82 and reduced the living cell number. The expression of a FAM83G phosphorylation-resistant mutant (S356A-FAM83G) had no effect on the living cell number or the induction of spontaneous apoptosis. By contrast, the introduction of a synthetic peptide encompassing FAM83G S356 into HCT116 and HepG2 cells decreased HSP27 S15 and S82 phosphorylation and induced spontaneous apoptosis. On the other hand, the introduction of FAM83G phosphorylation-resistant mutant synthesized peptides (S356A-AF-956 and S356A-AG-066) did not reduce the living cell number or induce spontaneous apoptosis. The endogenous expression of HSP27 and FAM83G was apparently greater in HCT116 and HepG2 cells compared with in CHO cells. In various types of lung cancer cell lines, the FAM83G messenger RNA (mRNA) level in non-small lung cancer cells was at a similar level to that in non-cancerous cells. However, the FAM83G mRNA level in the small cell lung cancer cell lines was variable, and the HSP27 mRNA level in FAM83G mRNA-rich types was greater than that in FAM83G mRNA-normal range types. Taken together, these data demonstrate that FAM83G S356 phosphorylation modulates HSP27 phosphorylation and apoptosis regulation and that HSP27 is a counterpart of FAM83G.

## 1. Introduction

The family with sequence similarity 83 (FAM83) protein family comprises eight known members: FAM83A–FAM83H [1,2]. Members of the FAM83 family share a conserved N terminal domain of approximately 300 amino acids (Pfam DUF1669 domain), which is homologous to the phospholipase D catalytic domain [1,2]. However, owing to the absence of essential catalytic histidine residues, this domain is unlikely to have phospholipase activity [3,4]. Of the FAM83 members identified to date, only FAM83G has a consensus motif for serine/threonine protein kinase D1/protein kinase C mu (PKD1/PKCμ) phosphorylation, which is located on serine residue 356 (S356). Therefore, FAM83G likely has a unique physiological function among the FAM83 protein family members.

Overall, FAM83G is conserved across vertebrates. A missense mutation in the FAM83G gene causes hyperkeratosis in dogs, canines, and humans [2,4,5], and FAM83G gene truncation has been reported to be responsible for the wooly mutation in mice [1]. Additionally, FAM83G has been reported to play a key role in palmoplantar keratoderma [1,2,4]. However, it has not yet been reported whether FAM83G has a role in apoptosis in cancer cells. In this paper, we observe that FAM83G can function as a novel spontaneous apoptosis inducer.

## 2. Results

### 2.1. Effect of FAM83G Overexpression on CHO Cells

The analysis of the primary amino acid sequence of FAM83G using Scansite (3 Beta; http://scansite3.mit.edu/#home) revealed the presence of a consensus PKD1/PKCμ phosphorylation site at S356 (YALVKAKSVDEIAKS). Therefore, we first tested whether or not the FAM83G protein coimmunoprecipitated with PKD1/PKCμ (Figure 1a). We found that endogenous FAM83G protein was co-immunoprecipitated with endogenous PKD1/PKCμ, as shown in the pCMV6 lane (used as the control sample). On the other hand, overexpressed FAM83G was co-immunoprecipitated with an increased amount of PKD1/PKCμ compared to that of pCMV6 samples as shown in the pCMV6-wild-type (WT)-FAM83G lane. These data suggested that FAM83G is associated with PKD1/PKCμ and, moreover, that PKD1/PKCμ possibly phosphorylates FAM83G. Furthermore, the degree of HSP27 phosphorylation at the S82 residue appeared to be inversely correlated with the FAM83G expression level (Figure 1b). As PKD1/PKCμ is known to phosphorylate HSP27 S82, these data support a model in which FAM83G interacts with PKD1/PKCμ and thereby prevents HSP27 phosphorylation via PKD1/PKCμ.

Next, we assessed the effect of FAM83G overexpression in CHO cells used as non-cancerous cells. CHO cells were used because we were able to reach 85–100% transfection efficiency by electroporation, as previously reported [6]. In this system, the amount of overexpressed FAM83G was typically ~10-fold greater than that of endogenous FAM83G (Figure 2a). To examine the role of FAM83G S356 phosphorylation on cell survival, we compared the effects of expressing WT-FAM83G, a Y586A-FAM83G mutant that maintains S356 phosphorylation, and a S356A-FAM83G mutant that is S356 phosphorylation-resistant. The expression of both WT-FAM83G and the Y586A-FAM83G mutant caused reductions in CHO cell survival (Figure 1c, columns 2 and 3, respectively). By contrast, we found a significant effect on CHO cell survival when the phosphorylation-defective S356A-FAM83G mutant was expressed in the CHO cells (Figure 1c, column 4). The cell survival of S356A-FAM83G-expressing cells was greater than that of pCMV6-introduced cells. This is probably due to overexpressed S356A-FAM83G canceling the endogenous FAM83G function too. We confirmed that the expression levels of the WT-FAM83G, Y586A-FAM83G, and S356A-FAM83G proteins in the CHO cells were similar (Figure 1d). Therefore, in combination, these data indicate that FAM83G S356 phosphorylation is necessary for FAM83G overexpression to reduce the live cell number.

To assess the function of FAM83G in cancer cells, WT-FAM83G was overexpressed in HCT116 cells (Figure 2a). We estimated that the amount of overexpressed WT-FAM83G was again ~10-fold greater than that of the endogenous FAM83G. When we overexpressed FAM83G, we found that HCT116 cells displayed a significantly reduced live cell amount similar to the CHO cells (compare Figure 1c with Figure 2b). However, there is a possibility that FAM83G possesses other phosphorylation sites besides S356. As such, those additional phosphorylation sites could also contribute to the reduction of live cell numbers upon FAM83G overexpression.

To determine whether S356 phosphorylation is important for WT-FAM83G’s effect on cell survival, we generated three FAMS83G peptides (AF-859, AF-956, and AG-066) that included the FAM83G S356 consensus site for PKD1/PKCμ-mediated phosphorylation. These peptides’ information is described in the “Materials and Methods” section. In short, all three peptides contained identical FAM83G sequences but either lacked an additional amino terminal antenna peptide for cell entry (AF-859; random peptide with missing antenna peptide as the control peptide), possessed an additional amino antenna peptide that was specific for entry into colon cancer cells (AF-956), or possessed an additional amino antennae peptide that was specific for entry into liver cancer cells (AG-066) [7].

First, we attempted to confirm that active PKD1/PKCμ kinase was able to phosphorylate AF-956 by using an in vitro kinase assay. As shown in Figure 3a, AF-956 was confirmed to be a substrate for PKD1/PKCμ kinase because PKD1/PKCμ directly phosphorylated AF-956.

Following the treatment of HCT116 cells for 36 h with 54 μM concentrations of each peptide, we observed that only AF-956 was able to cause a significant reduction in live cell number (Figure 3b) without affecting endogenous FAM83G protein levels (Figure 3c). Furthermore, the S15 and S82 phosphorylation of HSP27 was significantly reduced upon the treatment of the HCT116 cells with AF-956 but not upon AF-859 or AG-066 treatment (Figure 3d). Moreover, in contrast to AF-859 and AG-066, the AF-956 treatment of the HCT116 cells was observed to increase cellular DNA laddering (Figure 3e). Consistent with the increment in cellular DNA laddering, the AF-956 treatment also increased the levels of cleaved caspase 3 and PARP in the HCT116 cells (Figure 3f). However, the control AF-859 and AG-066 peptides had no effect on the cleavage levels of these proteins in the HCT116 cells (Figure 3f). Therefore, these data support our model wherein FAM83G S356 phosphorylation is required to induce spontaneous apoptosis in HCT116 cells.

### 2.2. A FAM83G S356A-AF-956 Peptide Cannot Activate Apoptosis in HCT116 Cells

To further determine whether S356 phosphorylation is important for WT-FAM83G for the reduction in cell survival of HCT116 cells, we generated an S356A-AF-956 peptide that is resistant to PKD1/PKCμ-mediated phosphorylation. The amino acid sequence of S356A-AF-956 is described in the “Materials and Methods” section. As shown in Figure 4a, following the treatment of HCT116 cells for 36 h with 54 μM S356A-AF-956 peptide, we observed that S356A-AF-956 had no significant effect, with no change in the levels of endogenous FAM83G protein (Figure 4b). In addition, the S15 and S82 phosphorylation of HSP27 was not changed upon the treatment of the HCT116 cells with S356A-AF-956 or AF-859 treatment (Figure 4c). Moreover, the S356A-AF-956 treatment of the HCT116 cells had no effect on cellular DNA laddering (Figure 4d). Consistent with the absence of apoptosis, the S356A-AF-956 peptide was unable to activate caspase-3 and PARP in HCT116 cells (Figure 4e).

Together, these data support our model wherein FAM83G S356 phosphorylation is required to induce spontaneous apoptosis in CHO and HCT116 cells.

### 2.3. The FAM83G AG-066 Peptide can Activate Apoptosis through a Decrease in HSP27 Phosphorylation in HepG2 Cells

Originally, AG-066 was designed to specifically target liver cancer cells rather than colon cancer cells. To confirm that AG-066 is biologically active, we examined whether the AG-066 treatment of HepG2 cells could modulate HSP27 phosphorylation and cause apoptosis. The treatment of HepG2 cells with 54 μM AG-066 for 72 h markedly reduced the live cell number compared with the AF-859 negative control (Figure 5a). Although the AG-066 treatment of the HepG2 cells had no effect on the endogenous FAM83G protein levels (Figure 5b), we did observe decreased HSP27 phosphorylation at S15 and S82 (Figure 5c). Moreover, in contrast to AF-859, AG-066 was observed to increase the levels of both cleaved caspase 3 and PARP in the HepG2 cells (Figure 5d). These data not only indicate that AG-066 is biologically active but also confirm that amino terminal antennae allow for the targeted delivery of peptides into specific cell types, as previously reported [7]. Therefore, in combination, our data show that FAM83G S356 phosphorylation is a critical step in the induction of spontaneous apoptosis in HepG2 cells too.

### 2.4. A FAM83G S356A-AG-066 Peptide Cannot Activate Apoptosis in HepG2 Cells

To further determine whether S356 phosphorylation is important for the WT-FAM83G regulation of cell number in HepG2 cells, we generated an S356A-AG-066 peptide whose S356 is resistant to PKD1/PKCμ-mediated phosphorylation. The amino acid sequence of S356A-AG-066 is described in the “Materials and Methods” section. As shown in Figure 6a, following the treatment of HepG2 cells for 72 h with 54 μM S356A-AG-066 peptide, we observed no significant change in the live cell number. The endogenous FAM83G protein levels was also unchanged (Figure 6b). In addition, the S15 and S82 phosphorylation of HSP27 was not changed upon treatment of the HepG2 cells with S356A-AG-066 or AF-859 treatment (Figure 6c). Consistent with the lack of induced apoptosis, the S356A-AG-066 treatment did not affect the levels of uncleaved caspase 3 and PARP in the HepG2 cells (Figure 6d). Thus, these data also support our model wherein FAM83G S356 phosphorylation is required to induce apoptosis in CHO, HCT116, and HepG2 cells.

### 2.5. Comparison of Endogenous FAM83G and HSP27 Protein Levels among CHO, HepG2, and HCT116 Cells

We studied whether there is a difference in endogenous FAM83G and HSP27 protein levels between non-cancerous and cancerous cells. As shown in Figure 7, the FAM83G protein level in cancerous cells (HCT116 and HepG2 cells) was apparently higher than that in non-cancerous cells (CHO cells). However, this conclusion remains tentative as the sequences of the human and hamster antibody epitopes are only 76% identical. Interestingly, when we looked at the HSP27 protein level, cancerous cells in which the FAM83G protein level was higher contained a higher amount of HSP27 protein compared with non-cancerous cells in which the FAM83G protein level was significantly lower. Thus, an association between endogenous FAM83G and HSP27 protein levels seems to exist. This probably occurs as a compensation to maintain tumor cell survival.

### 2.6. Comparison of FAM83G mRNA Level Estimated via qPCR in Various Lung Cancer Cell Lines

To further investigate the relationship between endogenous FAM83G and HSP27 protein levels in other cancerous cells, we estimated the *FAM83G* mRNA level via qPCR of five noncancerous cell lines, nine NSCLC cell lines with an *EGFR* mutation, four NSCLC cell lines with a *BRAF* mutation, 10 NSCLC cell lines with wild-type *EGFR*/*BRAF*/*KRAS*, 11 NSCLC cell lines with a *KRAS* mutation, and 18 SCLC cell lines with wild-type *EGFR*/*BRAF*/*KRAS*. The results were 1.13 ± 1.08 in the non-cancerous cells, 1.22 ± 1.01 in the NSCLC cells with an *EGFR* mutation, 0.69 ± 0.17 in the NSCLC cells with a *BRAF* mutation, 1.09 ± 1.09 in the NSCLC cells with wild-type *EGFR*/*BRAF*/*KRAS*, 0.72 ± 0.27 in the NSCLC cells with a *KRAS* mutation, and 19.42 ± 44.16 in the SCLC cells. 

As shown in Figure 8a, the *FAM83G* mRNA level in all of NSCLCs was almost equal to or lower than that in the non-cancerous cells. On the contrary, the *FAM83G* mRNA level in the SCLCs varied, and some of them showed a higher level compared with the non-cancerous cells. Therefore, we decided to compare the *HSP27* mRNA level between SCLCs with a lower *FAM83G* mRNA level compared with non-cancerous cells and SCLCs with a higher *FAM83G* mRNA level compared with non-cancerous cells.

As shown in Figure 8b, the *HSP27* mRNA level of SCLCs with a higher *FAM83G* mRNA level compared with non-cancerous cells was significantly higher than in SCLCs with a lower *FAM83G* mRNA level compared with non-cancerous cells. Thus, it was confirmed that FAM83G-rich cancer cells possess a large amount of HSP27 as a compensatory mechanism as described above.

## 3. Discussion

In this study, we revealed that the overexpression of WT-FAM83G in CHO cells significantly reduced the live cell number. We also demonstrated that the phosphorylation of the FAM83G S356 residue was required for the reduction of the live cell number, as the CHO cells were unaffected upon the overexpression of a FAM83G S356A mutant resistant to S356 phosphorylation. Although we did not directly identify the kinase(s) responsible for the phosphorylation of FAM83G S356, the most likely candidate is PKD1/PKCμ, because FAM83G was observed to associate with PKD1/PKCμ. In fact, an active form of PKD1/PKCμ could phosphorylate the FAM83G peptide, including the S356 portion.

Regarding how FAM83G reduced the cell number, PKD1/PKCμ has been reported to directly phosphorylate the S15 and S82 residues of HSP27, which is known to protect cells from apoptosis [8,9,10,11]. We revealed that the expression of either WT-FAM83G or the Y586A-FAM83G mutant in CHO cells significantly decreased the live cell number, while the expression of the S356A-FAM83G mutant that is S356 phosphorylation-resistant had no effect. On the basis of these findings, we postulated that FAM83G and HSP27 may compete to bind to PKD1/PKCμ, which would account for the ability of FAM83G peptides to mimic the effect of WT-FAM83G. Although we also demonstrated that the phosphorylation of the HSP27 S82 residue was inversely correlated with the FAM83G expression levels, further studies are required to determine whether FAM83G functions as a direct antagonist of PKD1/PKCμ-mediated HSP27 phosphorylation.

To determine whether a sequence encompassing the FAM83G S356 site is sufficient to induce apoptosis, we generated FAMS83G peptides that included the S356 FAM83G consensus site for PKD1/PKCμ-mediated phosphorylation. All peptides contained identical FAM83G sequences but either lacked an additional amino terminal antenna peptide for cell entry with a random peptide as control peptide or possessed an additional amino antenna peptide specific for entry into colon or liver cancer cells. After we confirmed that the designed peptides could be phosphorylated by active PKD1/PKCμ kinase, we treated HCT116 colon cancer cells and HepG2 liver cancer cells with those peptides. We observed a significant reduction in live cell numbers as well as a reduction in the S15 and S82 phosphorylation of HSP27. In addition, we observed increased cellular DNA laddering and cleaved caspase-3 and PARP levels in the HCT116 and HepG2 cells. Therefore, these data support our model wherein FAM83G S356 phosphorylation is a critical step in the induction of spontaneous apoptosis. In addition, to determine the physiological relevance of FAM83G in cancer cells, we measured and compared the *FAM83G* mRNA levels among various human lung cancer cell lines. We discovered that the *FAM83G* mRNA level in NSCLC cell lines was similar to or lower than that in non-cancerous cell lines. However, the *FAM83G* mRNA level in SCLC cell lines varied; some showed a huge amount of FAM83G mRNA, while others showed a similar or lower *FAM83G* mRNA level compared with the non-cancerous cell lines.

Thus, the majority of lung cancer cells had a lower *FAM83G* mRNA level, protecting them from apoptosis. Surprisingly, though, some of them survived in spite of a high *FAM83G* mRNA level, suggesting that they had a compensative mechanism. When we looked at the *HSP27* mRNA level in the SCLC cell lines, the FAM83G-rich cells contained more *HSP27* mRNA compared with those lacking FAM83G. Therefore, HSP27 seemed to be a counterpart of FAM83G that rescued the cancer cells from the induction of spontaneous apoptosis by FAM83G.

## 4. Materials and Methods

### 4.1. Reagents

Anti-FAM83G (catalog number; ab121750, ab186119), anti-phospho-(S15) heat shock protein 27 (HSP27) (ab76313), and anti-phospho-(S82) HSP27 (ab155987) antibodies, and the Apoptotic DNA Ladder Isolation Kit (ab65627) for cultured cells were purchased from Abcam (Cambridge, UK). Anti-poly (ADP-ribose) polymerase (anti-PARP) (9542), anti-caspase 3 (9662), anti-phospho-(S82) HSP27 (2401), anti-HSP27 (2402), and α-tubulin (2144) antibodies were purchased from Cell Signaling Technology (Danvers, MA, USA). Horseradish peroxidase-conjugated anti-rabbit (31460) and mouse (31437) IgG antibodies were purchased from Thermo Fisher Scientific (Waltham, MA, USA). Anti-FLAG M2 (F1804) antibody was purchased from Sigma-Aldrich (St. Louis, MO, USA). WT-FAM83G ORF Clone (MR210551) was purchased from OriGene (Rockville, MD, USA). The ACS^TM^ XTT Cell Proliferation Assay (4891-025-K) was purchased from TREVIGEN (Gaithersburg, MD, USA). The Live/Dead Cell Staining Kit II (PKCA707-3002) was purchased from PromoCell (Heidelberg, Germany). All cell culture media and reagents were purchased from Thermo Fisher Scientific. All the other chemicals used in this study were purchased from Sigma-Aldrich. The Peptide Institute (Osaka, Japan) synthesized the following oligopeptides: AF-859 (random peptide with missing antenna peptide), KASKEADAIVSLVK; AF-956, DSLKSYWYLQKFSWRYALVKAKS356VDEIAKS; and AG-066, KRPTMRFRYTWNPMKYALVKAKS356VDEIAKS. GenScript (Piscataway, NJ, USA) synthesized the following oligopeptides: S356A-AF-956, DSLKSYWYLQKFSWRYALVKAKAVDEIAKS; and S356A-AG-066, KRPTMRFRYTWNPMKYALVKAKAVDEIAKS.

### 4.2. Cell Culture

Chinese hamster ovary (CHO) cells were maintained in Eagle’s Minimum Essential Medium supplemented with 10% (*v*/*v*) fetal bovine serum [6]. HCT116 and HepG2 cells were purchased from the Japanese Collection of Research Bioresources Cell Bank (Osaka, Japan); these cells were maintained in Dulbecco’s Modified Eagle Medium supplemented with 10% (*v*/*v*) fetal bovine serum [8]. The cells were grown to confluency and incubated with either AF-859, AF-956, AG-066, S356A-AF-956, or S356A-AG-066 at the concentrations described in the figure legends at 37 °C for 24–72 h. All peptides were resuspended in 0.1% (*v*/*v*) trifluoroacetic acid. We used a total of five non-cancerous human bronchial epithelial cell lines (NHBE, SAEC, BEAS2B, HBEC3, and HBEC4), nine non-small cell lung cancer (NSCLC) cell lines with an epidermal growth factor receptor (*EGFR*) mutation (*EGFR* exon 19 deletion (HCC2935, HCC4006, PC9, HCC2279, and HCC827), *EGFR* L858R mutation (H3255 and HCC4011), *EGFR* L858R/T790M mutation (H1975), and *EGFR* exon 19 deletion/T790M (H820)), four NSCLC cell lines with a *BRAF* mutation (*BRAF* G466V (H1666), *BRAF* G469A (H1755), *BRAF* G469V (H1395), and *BRAF* L597V (H2087)), eleven NSCLC cell lines with a *KRAS* mutation (*KRAS* G12C (H2122, HCC44, H1792, HCC4017, and H358), *KRAS* G12V (H441), *KRAS* G13D (HCC515), *KRAS* G12R (H1264 and H157), *KRAS* G12A (H2009), and *KRAS* Q61H (H460)), eighteen small cell lung cancer (SCLC) cell lines with wild-type *EGFR*/*BRAF*/*KRAS* (H187, H345, H378, H524, H526, H740, H889, H1045, H1092, H1184, H1238, H1339, H1607, H1618, H1672, H1963, H2171, and H2227), and ten NSCLC cell lines with wild-type *EGFR*/*BRAF*/*KRAS* (H1299, H1819, HCC95, H838, H1437, H661, HCC15, HCC78, H1648, and HCC193) [5].

### 4.3. Quantitative Transient Transfection by Electroporation

CHO cells were suspended in 500 μL of PBS with the empty pCMV6 vector or the pCMV6 mammalian expression vector containing the mouse origin wild-type FAM83G cDNA, which was purchased from OriGene Technologies (Rockville, MD, USA; catalog no: MR210551). According to BLAST and using the protein sequence, the identities between the *Homo sapiens* FAM83G and the *Mus musculus* FAM83G were 80% (659/823), the positives were 85% (700/823), and the gaps were 1% (1011/823). After suspension, the CHO cells were then electroporated at 340 V and 960 μF and then plated in α-minimal essential medium containing 10% (*v*/*v*) fetal bovine serum. Cell debris was removed by replacing existing media with fresh media after 12 h and then 30 h. After 48 h, the transfected cells were used for experiments [6].

### 4.4. Evaluation of mRNA Levels via qPCR

The *FAM83G* mRNA levels were analyzed via RT-qPCR as previously described [8]. Primers and probes for *FAM83G* and HSP27 (*HSPB1*) were obtained from OriGene Technologies (NM_178618 and NM_001540, respectively), whereas those for glyceraldehyde 3-phosphate dehydrogenase (*GAPDH*) were obtained from Applied Biosystems (Assay ID: Hs99999905_m1; Tokyo, Japan). To normalize the amount of input cDNA, a quantitative analysis was performed using *GAPDH* as an internal reference. Relative expression values were computed using the comparative cycle threshold method.

### 4.5. Immunoprecipitation

Immunoprecipitations were performed by incubating 2 mg of cell extracts with 4 μg of a FAM83G antibody at 4 °C for 2 h. The samples were then incubated with protein A-sepharose at 4 °C for 1 h. Subsequently, the immunoprecipitated samples and whole cell lysates were resuspended in a sodium dodecyl sulfate (SDS) sample buffer (125 mM Tris-HCl pH 6.8, 20% (*v*/*v*) glycerol, 4% (*w*/*v*) SDS, 100 mM dithiothreitol, and 0.1% (*w*/*v*) bromophenol blue) and heated at 100 °C for 5 min. Finally, the samples were separated via SDS polyacrylamide gel electrophoresis (SDS-PAGE) [8,12].

### 4.6. Immunoblotting

Scraped frozen cells were rocked at 4 °C for 10 min with NP-40 (IGEPAL) lysis buffer (25 mM HEPES pH 7.4, 10% (*v*/*v*) glycerol, 50 mM sodium fluoride, 10 mM sodium pyrophosphate, 137 mM sodium chloride, 1 mM sodium orthovanadate, 1 mM phenylmethylsulfonyl fluoride, 10 μg/mL of aprotinin, 1 μg/mL of pepstatin, and 5 μg/mL of leupeptin). Insoluble material was separated from the soluble extract by centrifugation (15,000 rpm at 4 °C for 10 min), and the total protein concentration in the supernatant was determined using the bicinchoninic acid method. The total protein content in each sample was normalized prior to subsequent analysis. The samples were resuspended in SDS sample buffer and heated for 5 min at 100 °C. They were then separated by SDS-PAGE and electrophoretically transferred to polyvinylidene difluoride membranes; finally, they were immunoblotted with specific antibodies as indicated in the figure legends [6,8].

### 4.7. Site-Directed Mutagenesis

The Y586A-FAM83G and S356A-FAM83G mutants were created using the QuikChange Site-Directed Mutagenesis Kit (Agilent, Santa Clara, CA, USA) according to the manufacturer’s instructions.

### 4.8. Evaluation of Apoptosis Using the DNA Ladder Method

The DNA ladder method was performed according to the manufacturer’s instructions. Briefly, cells were lysed with Tris-EDTA lysis buffer and mixed with the enzyme solution supplied in the kit. Following the isolation and purification of the DNA, samples were analyzed by agarose gel electrophoresis, and ethidium bromide-stained DNA was visualized by ultraviolet transillumination.

### 4.9. In Vitro Synthetic Peptide (AF-956) Phosphorylation

Active PKD1/PKCμ was purchased from SignalChem (Richmond, BC, Canada), and 200 mU of the recombinant active form of PKD1/PKCμ was incubated with 2.5 μM of synthesized oligopeptides and kinase reaction buffer. The kinase reaction was then stopped by the addition of ice-cold TCA, and the solution was spotted onto phosphocellulose paper. This paper was extensively washed before a scintillation cocktail was added to it; subsequently, the radiation activity was measured [13]. As a control, AF-859 (random peptide) was used.

### 4.10. Live Assay

The Live/Dead cell assay was performed according to the manufacturer’s instructions. Briefly, cells were initially grown in well of a 96-well microplate. After washing the cells in a serum-free buffer, 1 μM Calcein-AM was added to each well. The wells were then incubated at room temperature for 30–45 min and were finally analyzed by measuring fluorescence using a microplate reader.

### 4.11. Statistical Analysis

All data in the figures are expressed as the mean ± standard deviation. One-factor ANOVA was used to compare the mean values of all the groups. The Tukey–Kramer multiple comparisons procedure was used to determine the statistical differences between the means, with *p* < 0.05 deemed to be statistically significant. The analyses were conducted using the InStat 2.00 program.

## 5. Conclusions

This study discovered that FAM83G functions as a spontaneous apoptosis inducer by decreasing the phosphorylation of HSP27 at S15 and S82. In FAM83G-rich cancer cells, the cells increased the *HSP27* mRNA or HSP27 protein levels and tried to rescue themselves from apoptosis induction via reducing FAM83G S356 phosphorylation. Moreover, our investigation strongly suggests that FAM83G-derived peptides should be investigated further as promising lead compounds for the treatment of some types of cancers, especially those whose FAM83G protein level is lower compared with that of non-cancerous cells.

## Figures and Tables

**Figure 1 molecules-25-02810-f001:**
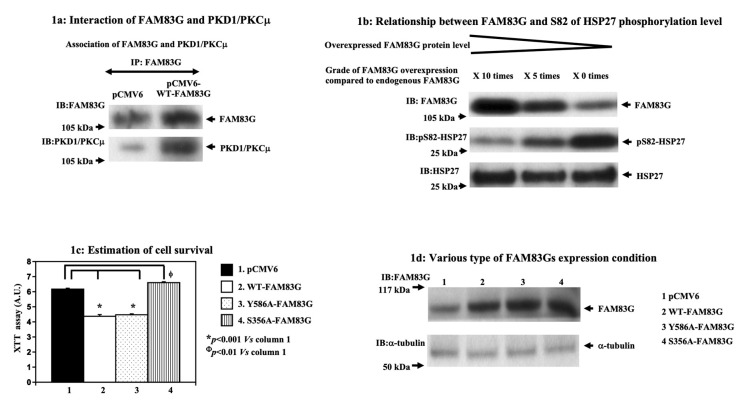
Effect of overexpressed FAM83G with or without S356 phosphorylation on cell numbers. (**a**) Interaction between FAM83G and PKD1/PKCμ; pCMV6 and pCMV6-wild type (WT)-FAM83G were expressed in CHO cells, FAM83G was immunoprecipitated, and then the immunoprecipitated FAM83G was immunoblotted with an antibody specific for FAM83G and PKD1/PKCμ. Endogenous FAM83G was co-immunoprecipitated with endogenous PKD1/PKCμ (pCMV6 lane, control sample). Overexpressed FAM83G was co-immunoprecipitated with increased PKD1/PKCμ levels (pCMV6-WT-FAM83G lane). The experiments were independently performed in triplicate. (**b**) Phosphorylation of the HSP27 S82 residue is inversely correlated with the levels of FAM83G expression. Upper panel: exogenously expressed FAM83G and endogenous FAM83G protein levels in CHO cells following immunoblotting (IB) with a FAM83G-specific antibody. Middle panel: the relative levels of S82-phosphorylated HSP27 are shown by IB with a specific anti-phospho-(S82) HSP27 antibody. Lower panel: relative HSP27 protein levels following IB with a specific anti-phospho-(S82) HSP27 antibody (strip and re-probe technique). The experiments were independently performed in triplicate. (**c**) Cell survival following the expression of pCMV6, WT-FAM83G, Y586A-FAM83G, and S356A-FAM83G in CHO cells; cell survival (relative number of live cells), as measured by the XTT assay. An empty pCMV6 plasmid transfected into the CHO cells acted as the control. The results of the statistical analyses, which are shown to the right of the bar graph, indicated that phosphorylation at S356 was required for WT-FAM83G overexpression to significantly reduce live cell numbers. The experiments were independently performed in triplicate. (**d**) Expression levels of WT-FAM83G and the FAM83G mutants; exogenously expressed FAM83G protein levels were approximately 7-fold greater than the endogenous FAM83G level. The relative levels of endogenous-tubulin are shown as loading controls. The experiments were independently performed in triplicate.

**Figure 2 molecules-25-02810-f002:**
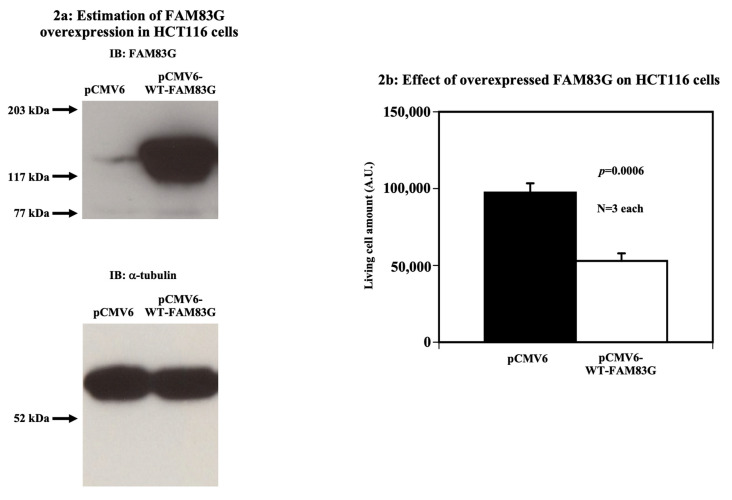
Effect of FAM83G overexpression on HCT116 cells. (**a**) Estimation of FAM83G overexpression in HCT116 cells. The amount of overexpressed wild type (WT)-FAM83G in HCT116 cells was ~10-fold greater than that of endogenous FAM83G (upper panel). α-tubulin blotting proved that equal protein amounts were loaded in each sample (lower panel). The experiments were independently performed in triplicate. (**b**) Overexpressed WT-FAM83G reduces the number of live HCT116 cells; relative live HCT116 cell numbers were quantified using the LIVE/DEAD cell assay following the expression of an empty pCMV6 vector (control) or pCMV6-WT-FAM83G. The results of the statistical analyses are shown: WT-FAM83G overexpression significantly reduced the number of live cells. The experiments were independently performed in triplicate.

**Figure 3 molecules-25-02810-f003:**
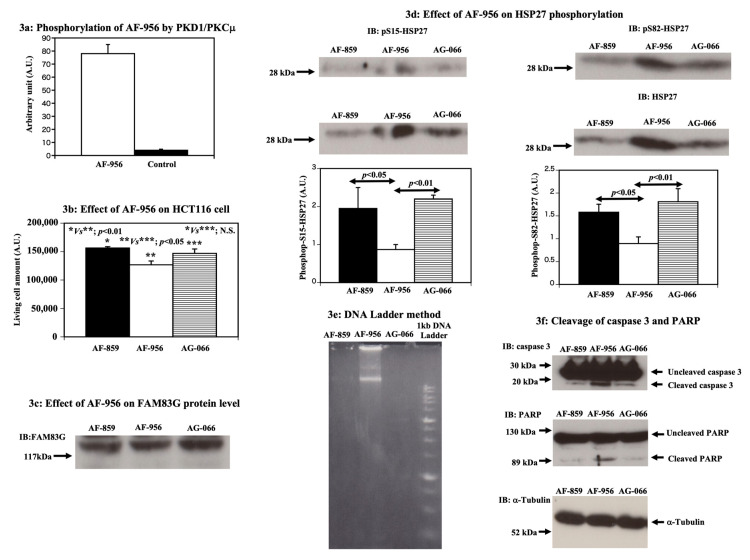
The FAM83G AF-956 peptide activates apoptosis through the reduction of HSP27 phosphorylation in HCT116 cells. (**a**) Phosphorylation of AF-956 by PKD1/PKCμ, An in vitro kinase assay showed that PKD1/PKCμ phosphorylated AF-956. The AF-859 peptide (random peptide) was used as a control. The experiments were independently performed in triplicate. (**b**) AF-956 reduces the live HCT116 cell number, Treatment with 54 μM AF-956 significantly reduced the live cell number (middle column); treatment with 54 μM AF-859 (left column) or 54 μM AG-066 (right column) had no significant effect on cell number. The experiments were independently performed in triplicate. (**c**) AF-956 has no effect on FAM83G protein levels, Immunoblots showing that the peptides AF-859, AF-956, and AG-066 had no effect on endogenous FAM83G protein levels, experiments independently performed in triplicate. (**d**) AF-956 reduces the S15 and S82 phosphorylation of HSP27. Whole cell lysates from HCT116 cells treated with AF-859, AF-956, or AG-066 were analyzed by immunoblotting (IB) with antibodies specific for anti-phospho-(S15) HSP27 (middle panel, left) and anti-phospho-(S82) HSP27 (middle panel, right). IB with an HSP27-specific antibody: AF-956 treatments increased HSP27 protein levels in HCT116 cells relative to AF-859 and AG-066 treatments (top panels, left and right). Bottom panels (bar graphs) show the S15 and S82 phosphorylation of HSP27, i.e., the intensity of the phospho-S15 HSP27 and phosphor-S82 HSP27 bands relative to that of the HSP27 band. The AF-956 treatment (middle bar) of HCT116 cells significantly reduced both the S15 and S82 phosphorylation of HSP27, whereas AF-859 (left bar) and AG-066 (right bar) treatments had no effect. The experiments were independently performed in triplicate. (**e**) AF-956 causes apoptosis in HCT116 cells (DNA Ladder analysis). Apoptotic DNA fragmentation was qualitatively analyzed using the DNA Ladder method. Lanes, from left to right: AF-859-treated cells, AF-956-treated cells, AG-066-treated cells, 1 kb DNA ladder marker. Analysis showed that AF-956-treated cells caused apoptosis. The experiments were independently performed in triplicate. (**f**) AF-956 causes apoptosis in HCT116 cells (cleavage of caspase 3 and PARP proteins). HCT116 cells treated with AF-956 showed increased caspase 3 and PARP protein cleavages. Top panel: levels of cleaved and uncleaved caspase 3 proteins (arrows) following the treatment of HCT116 cells with AF-859 (left lane), AF-956 (middle lane), and AG-066 (right lane). Middle panel: levels of cleaved and uncleaved PARP proteins (arrows) following the treatment of HCT116 cells with AF-859 (left lane), AF-956 (middle lane), and AG-066 (right lane). Bottom panel: α-tubulin IB shows equivalent protein loading in each lane. The experiments were independently performed in triplicate.

**Figure 4 molecules-25-02810-f004:**
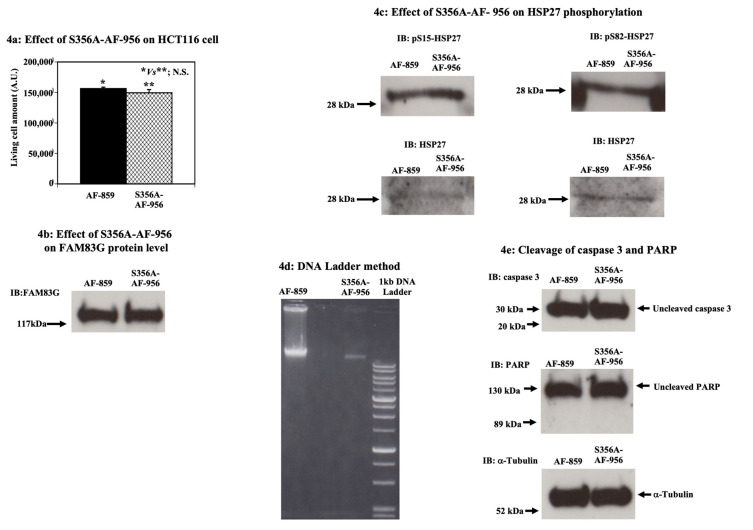
The effect of the FAM83G S356A-AF-956 peptide on apoptosis in HCT116 cells. (**a**) Effect of S356A-AF-956 on HCT116 cells. The effects of S356A-AF-956 (right column) and AF-859 (left column) peptide treatment (both 54 µM) on HCT116 cells were not significantly different. The experiments were independently performed in triplicate. (**b**) Effect of S356A-AF-956 on FAM83G protein levels. Immunoblots show that the peptides AF-859 and S356A-AF-956 had no effect on the endogenous FAM83G protein levels. The experiments were independently performed in triplicate. (**c**) Effect of S356A-AF-956 on HSP27 phosphorylation. Whole cell lysates from HCT116 cells treated with either AF-859 or S356A-AF-956 were analyzed by immunoblotting (IB) with antibodies specific for anti-phospho-(S15) HSP27 (top panel, left) and anti-phospho-(S82) HSP27 (top panel, right). IB with an HSP27-specific antibody: AF-859 and S356A-AF-956 treatments had similar effects on HSP27 protein levels in HCT116 cells; thus, S356A-AF-956 treatment did not affect S15 and S82 phosphorylation. The experiments were independently performed in triplicate. (**d**) S356A-AF-956 did not cause apoptosis in HCT116 cells (DNA Ladder analysis). Apoptotic DNA fragmentation was qualitatively analyzed using the DNA Ladder method. Lanes, left to right: AF-859-treated cells, S356A-AF-956-treated cells, and 1 kb DNA ladder marker. Analysis showed that S356A-AF-956-treated cells did not cause apoptosis. The experiments were independently performed in triplicate. (**e**) S356A-AF-956 did not cause apoptosis in HCT116 cells (cleavage of caspase 3 and PARP proteins). Cells treated with S356A-AF-956 showed no change in both caspase 3 and PARP protein cleavages. Top panel: levels of uncleaved caspase 3 proteins (arrow) following the treatment of HCT116 cells with AF-859 (left lane) and S356A-AF-956 (right lane). Middle panel: levels of uncleaved PARP proteins (arrow) following the treatment of HCT116 cells with AF-859 (left lane) and S356A-AF-956 (right lane). Bottom panel: α-tubulin IB shows equivalent protein loading in each lane. The experiments were independently performed in triplicate.

**Figure 5 molecules-25-02810-f005:**
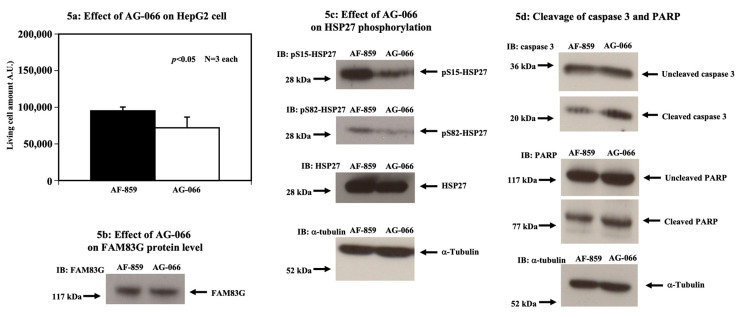
The FAM83G AG-066 peptide activates apoptosis through the reduction of HSP27 phosphorylation in HepG2 cells. (**a**) AG-066 reduces the live HepG2 cell number. The treatment of HepG2 cells with 54 μM AG-066, containing an amino terminal antenna peptide allowing for specific entry into live cancer cells (right column), significantly reduced the number of live HepG2 cells relative to the effect of treatment with 54 μM AF-859, which lacked an amino terminal antenna peptide (left column). The experiments were independently performed in triplicate. (**b**) AG-066 does not affect FAM83G protein levels. Immunoblots show that the peptides AF-859 and AG-066 had no effect on the endogenous FAM83G protein levels. The experiments were independently performed in triplicate. (**c**) Effect of AG-066 on HSP27 phosphorylation. Whole cell lysates of cells treated with either AF-859 (left lane) or AG-066 (right lane) were analyzed by immunoblotting (IB) with antibodies specific for anti-phospho-(S15) HSP27 (top panel) and anti-phospho-(S82) HSP27 (second panel down). HSP27 IB shows AG-066 had no effect on endogenous HSP27 protein levels relative to AF-859-treatment (third panel down). α-tubulin IB shows that sample loading was equivalent in all of the lanes (bottom panel). These data indicate that AG-066 treatment reduces both the S15 and S82 phosphorylation of HSP27 in HepG2 cells. The experiments were independently performed in triplicate. (**d**) AG-066 causes the apoptosis of HepG2 cells (cleavage of caspase 3 and PARP proteins). Cells treated with AG-066, but not AF-859, showed an increase in both caspase 3 and PARP protein cleavages. Top two panels: levels of cleaved and uncleaved caspase 3 proteins (arrow) following the treatment of HepG2 cells with AF-859 (left lane) and AG-066 (right lane). Third and fourth panels from the top: levels of cleaved and uncleaved PARP proteins (arrow) following the treatment of HepG2 cells with AF-859 (left lane) and AG-066 (right lane). Bottom panel: α-tubulin IB shows equivalent protein loading in each lane. The experiments were independently performed in triplicate.

**Figure 6 molecules-25-02810-f006:**
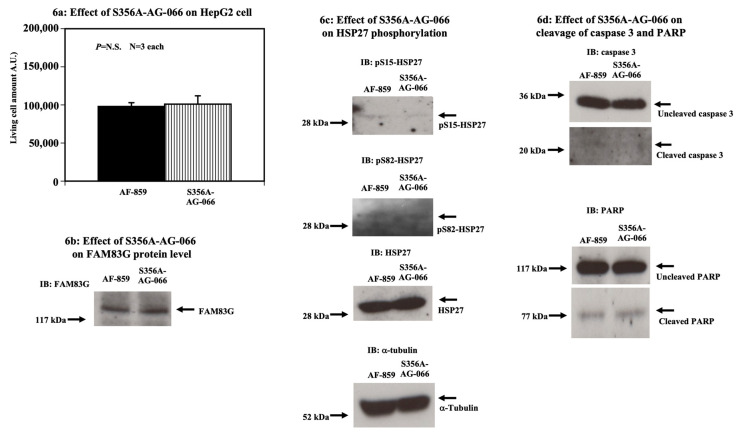
The effect of the FAM83G S356A-AG-066 peptide on apoptosis in HepG2 cells. (**a**) S356A-AG-066 does not affect the live HepG2 cell number. The effects of S356A-AG-066 (right column) and AF-859 (left column) (both 54 μM) on live HepG2 cells numbers were not significantly different. The experiments were independently performed in triplicate. (**b**) S356A-AG-0666 does not affect FAM83G protein levels. Immunoblots show that the peptides AF-859 and S356A-AG-066 had no effect on endogenous FAM83G protein levels. The experiments were independently performed in triplicate. (**c**) Effect of S356A-AG-066 on HSP27 phosphorylation. Whole cell lysates from HepG2 cells treated with either AF-859 or S356A-AG-066 were analyzed by immunoblotting (IB) with antibodies specific for anti-phospho-(S15) HSP27 (top panel, left) and anti-phospho-(S82) HSP27 (second panel down). IB with an HSP27-specific antibody showed that the S356A-AG-066 treatment of HepG2 cells did not affect HSP27 protein levels relative to AF-859 (third panel down); thus, S356A-AG-066 did not affect S15 and S82 phosphorylation in HepG2 cells. The experiments were independently performed in triplicate. (**d**) S356A-AG-066 does not cause apoptosis in HepG2 cells (cleavage of caspase 3 and PARP proteins). HepG2 cells treated with S356A-AG-066 showed no change in both caspase 3 and PARP protein cleavage relative to AF-859-treated HepG2 cells. Top two panels: levels of cleaved (second panel down) and uncleaved (top panel) caspase 3 proteins (arrow) following the treatment of HepG2 cells with AF-859 (left lane) and S356A-AG-066 (right lane). Bottom two panels: levels of cleaved (second panel down) and uncleaved (top panel) PARP proteins (arrows) following the treatment of HepG2 cells with AF-859 (left lane) and S356A-AG-066 (right lane). The positions of the uncleaved and cleaved PARP proteins are shown by arrows (right side). Bottom panel: α-tubulin immunoblotting shows equivalent protein loading in each lane (strip and re-probe technique was applied). The experiments were independently performed in triplicate.

**Figure 7 molecules-25-02810-f007:**
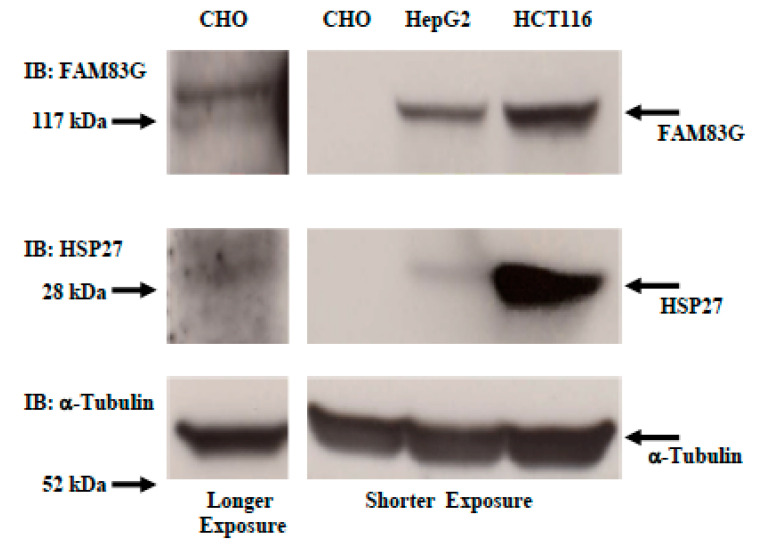
Comparison of endogenous FAM83G and HSP27 protein levels among CHO, HepG2, and HCT116 cells. Top panel: endogenous FAM83G protein. Second panel: endogenous HSP27 protein level. Bottom panel: -tubulin immunoblotting shows equivalent protein loading in each lane. Left lane: longer exposure to identify endogenous FAM83G and HSP27 proteins in CHO cells. Right panel: shorter exposure to identify endogenous FAM83G and HSP27 proteins in HepG2 and HCT116 cells. CHO; CHO cells.

**Figure 8 molecules-25-02810-f008:**
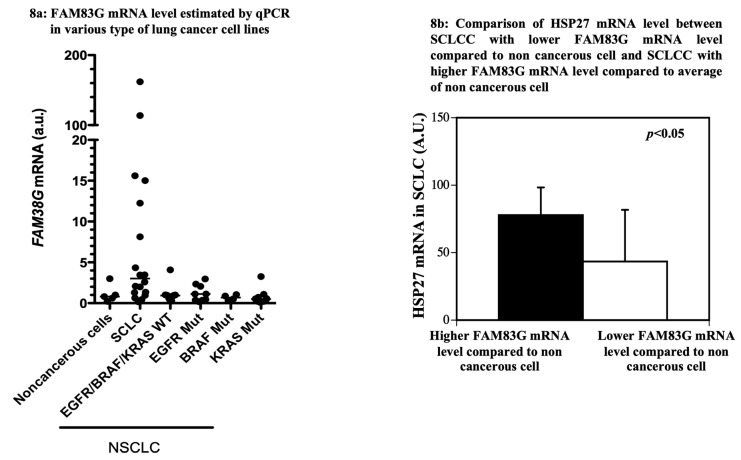
Estimation of FAM83G and HSP27 mRNAs in various lung cancer cell lines. (**a**) *FAM83G* mRNA levels estimated by qPCR in various lung cancer cell lines. Each closed circle shows each cell line’s *FAM83G* mRNA level. The *y*-axis represents the *FAM83G* mRNA levels; the *x*-axis describes each cell line type. The horizontal bar represents the median of *FAM83G* mRNA levels (a.u., arbitrary unit). Non-cancerous cells were used as controls (five cell lines). Lung cancer cell lines: SCLC, small cell lung cancer cells with wild-type EGFR/BRAF/KRAS (eighteen cell lines); EGFR/BRAF/KRAS WT, lung cancer cells with wild type EGFR/BRAF/KRAS (ten cell lines); EGFR Mut, lung cancer cells with EGFR mutation (nine cell lines); BRAF Mut, lung cancer cells with BRAF mutation (four cell lines); KRAS Mut, lung cancer cells with KRAS mutation (11 cell lines). (**b**) Comparison of *HSP27* mRNA levels between small cell lung cancer cells (SCLCCs) with lower *FAM83G* mRNA levels relative to non-cancerous cells and SCLCCs with higher *FAM83G* mRNA levels relative to non-cancerous cells. *HSP27* mRNA levels in cells with higher *FAM83G* mRNA levels relative to non-cancerous cells were significantly higher than those of cells with lower *FAM83G* mRNA levels relative to non-cancerous cells.

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
