# Peer review of "FAM83G Is a Novel Inducer of Apoptosis"

_molecules, 2020, doi:10.3390/molecules25122810_

Round 1

Reviewer 1 Report

In the manuscript, Okada and co-workers attempt to characterize the pro-apoptotic function of FAM83G protein. The manuscript is written in a good English, although profound double-checking is require to catch all mistakes (e.g. what is the STAR method?, what is the ‘vertical 25 striped column’?). Of course, this does not make the submitted material wrong. But what is more important, the manuscript described the experiments which seem to be designed in somewhat wrong way (e.g. the choice of human/hamster cells) and lack multiple necessary controls. Below is the list of the crucial issues that require addressing before accepting the manuscript for publication.

  1. Materials and methods: please specify the antibodies used in the study by providing catalogue numbers or clone names.
  2. Line 43, line 61: what do the superscript numbers stand for?
  3. Figure 1: Why was not a human but hamster (CHO) cell line used for the experiments? Was the overexpressed FLAG-FAM83G protein also of the chamster origin? Please describe this and the transfection procedure in the Materials and Methods section.
  4. Figure 1a: Important controls are missing. In addition to the provided data, for or the co-IP experiments it is necessary to show IB results for control IP samples without FAM83G overexpression (to prove for PKD1 binding to FAM83G, and not to, for example, the resin).
  5. Figure 1c: given the extremely low error bars, please provide with the information on the number of independent (performed separately) experiments. Please also provide the data from these experiments for the review process.
  6. Figure 1c: Does the graph really show relative values? The axis label says “U.”.
  7. Figure 1c: XTT does not directly measure “cell amounts” or “cell numbers”. Please correct.
  8. Figure 1c: The authors say: “we found no significant 142 effect on cell number or total cellular protein content when the phosphorylation-defective S356A143 FAM83G mutant was expressed”, while the graph shows significance at the level <0.01.
  9. Figure 2: Was the WT-FAM83G protein of the human origin for the overexpression in HCT116 cells? Please describe all the constructs in the Materials and Methods section.
  10. Figure 2: Please provide WB results for the successful overexpression of FAM83G in HCT116 cells.
  11. Figure 2, Figure 3, Figure 4b: The measurement of total protein in cell lysetes is not sufficient for judging on the cell survival. Please provide additional data, such as from the XTT experiment.
  12. Figure 2, Figure 3: Does the graph really show relative values? The axis label says “mg”.
  13. Figure 3: Important controls are missing. The result does not give the answer to the question whether it was the peptide or just electroporation that reduced the whole cell lysate protein content. Additionally, the experiment with scrambled (or random or mutant) peptide/peptides is required.
  14. Figure 3: The design of the experiment seem to be inadequate and lacks the reasoning. Why should the short peptide have the function of the whole FAM83G protein? Please explain e.g. by providing reasoning based on protein domains/interactions etc.
  15. Figure 4a: What was the vehicle control?
  16. Figure 4b: The untreated control is missing. Without that, one cannot say “treatment with 54 μM AF-859 (closed column) or 54 μM AG-066 (vertical 25 striped column) had no significant effect on the cell number.”
  17. Figure 4c: Immunoblot seems to be overexposed, lacks the untreated sample, and lacks loading control.
  18. Figure 4d: Please explain the reason why the overexpression of FLAG-FAM83G protein does not change the level of HSP27 expression (Figure 1b), and the treatment with AF-956 peptide gives such a HSP27 protein level increase.
  19. Figure 4d: Loading control is missing.
  20. Figure 4d: Untreated control is missing.
  21. Figure 4e, 4f: The experiments require additional control with scrambled (or random or mutant) peptide/peptides containing similar colon cancer entry signal as A-956 to prove for the pro-apoptotic activity of this particular peptide.
  22. Figure 5a: The untreated control is missing.
  23. Figure 5b: Immunoblot lacks the untreated sample, and lacks loading control.
  24. Figure 5c: Untreated control is missing.
  25. Figure 5d: The experiments require additional control with scrambled (or random or mutant) peptide/peptides containing similar liver cancer entry signal as AG-066 to prove for the pro-apoptotic activity of this particular peptide.
  26. Line 294-296: The statement ‘FAM83G protein level in cancerous cells (HCT116 and HepG2 cells) was apparently higher than that in noncancerous cells (CHO cells) cannot be acknowledged. HCT116 and HepG2 cells are human cells, while CHO are chamster cells. One cannot compare relative expression levels of the proteins from different species using WB technique without proving, that in all the used species the antibody binds to the targeted proteins with similar affinities.
  27. Figure 7b, lines 338-339: ‘Thus, it was confirmed that FAM83G-rich cancer cells possess a large amount of HSP27’ – this does not seem true. The figure shows that when FAM83G mRNA level is ‘smaller’, HSP27 mRNA is high, and when FAM83G mRNA is ‘larger’, HSP27 mRNA is lower. Therefore, there is rather a negative correlation, if there is any.

Author Response

Reviewer 1

Comments and Suggestions for Authors

In the manuscript, Okada and co-workers attempt to characterize the pro-apoptotic function of FAM83G protein. The manuscript is written in a good English, although profound doublechecking is require to catch all mistakes (e.g. what is the STAR method? what is the ‘vertical 25 striped column’?). Of course, this does not make the submitted material wrong. But what is more important, the manuscript described the experiments which seem to be designed in somewhat wrong way (e.g. the choice of human/hamster cells) and lack multiple necessary controls. Below is the list of the crucial issues that require addressing before accepting the manuscript for publication.

  1. Materials and methods: please specify the antibodies used in the study by providing catalogue numbers or clone names.

We have now provided the catalog number and company name for each antibody used in this study.

  1. Line 43, line 61: what do the superscript numbers stand for?

This was a typographical error and has been corrected.

  1. Figure 1: Why was not a human but hamster (CHO) cell line used for the experiments? Was the overexpressed FLAG-FAM83G protein also of the hamster origin? Please describe this and the transfection procedure in the Materials and Methods section.

As reviewer requested, we added the information describing the basis for the use CHO cells. These cells have a very high efficiency of transfection as we have previous reported [7].

  1. Figure 1a: Important controls are missing. In addition to the provided data, for or the co-IP experiments it is necessary to show IB results for control IP samples without FAM83G overexpression (to prove for PKD1 binding to FAM83G, and not to, for example, the resin).

We apologize for not clearly labeling the lanes of this figure. The samples without FAM3G expression (controls) are indicated as pCMV6 and those with FAM83G overexpression are indicated as pCMV6-WT-FAM83G. We have also clarified this in the text.

     5.Figure 1c: given the extremely low error bars, please provide with the information on the number of independent (performed separately) experiments. Please also provide the data from these experiments for the review process.

We performed those experiments three independent times that is now indicated. The numerical data is pCMV6; 6.1+0.06, pCMV6-WT-FAM83G; 4.5+0.05, pCMV6+Y356A-FAM83G; 4.6+0.04, pCMV6-S356A-FAM83G; 6.6+0.07.

    6.Figure 1c: Does the graph really show relative values? The axis label says “U.”

The y-axis has now been correct to A.U. for Arbitrary Units. The XTT assay is a colorimetric assay of cell survival. The absorbance values represent the cellular redox potential. As such these values are arbitrary measurements that need to be compared to the control cells.

     7.Figure 1c: XTT does not directly measure “cell amounts” or “cell numbers”. Please correct.

The reviewer is correct. We now state this as “cell survival” in line 15 on page 9. Also Figure 1c’s title is corrected as “Estimation of cell survival”.

     8.Figure 1c: The authors say: “we found no significant effect on cell number or total cellular protein content when the phosphorylation-defective S356AFAM83G mutant was expressed”, while the graph shows significance at the level <0.01.

The reviewer is absolutely right and thank you very much for your very critical comment. We are sorry about our mistake. We rewrote and added new sentence as “In contrast, we found significant effect on CHO cell survival when the phosphorylation-defective S356A-FAM83G mutant was expressed in the CHO cells (Figures 1c, column 4). The cell survival of S356A-FAM83G was bigger than that of pCMV6 introduced cells. This is probably due to overexpressed S356A-FAM83G canceled endogenous FAM83G function too.”

     9.Figure 2: Was the WT-FAM83G protein of the human origin for the overexpression in HCT116 cells? Please describe all the constructs in the Materials and Methods section.

The WT-FAM83G ORF Clone was from the mouse origin (MR210551) and was purchased from OriGene (Rockville, MD, USA). The identities between the homo sapiens FAM83G and the mus musculus FAM83G were 80 % (659/823), positives were 85 % (700/823), and the gaps were 1 % (11/823). The results are from the protein sequence using BLAST. We have described the ORF Clone in the Material and Methods line 19 on page 5-line 4 on page 6.

      10.Figure 2: Please provide WB results for the successful overexpression of FAM83G in HCT116 cells.

These data are now shown in Figure 2a as a typical result.

      11.Figure 2, Figure 3, Figure 4b: The measurement of total protein in cell lysates is not sufficient for judging on the cell survival. Please provide additional data, such as from the XTT experiment.

As requested, we have changed the total protein results to Live assay results, which assesses cell viability using Calcein-AM (PK-CA707-3002), which was purchased from PromoCell (Heidelberg, Germany). Please see response to criticism 11 too.

     12.Figure 2, Figure 3: Does the graph really show relative values? The axis label says “mg”.

We repeated all of experiment by living cell assay as described in Materials and Methods section. These new data are now represented in Figure 2b, 3b, 4a, 5a, and 6a.

     13.Figure 3: Important controls are missing. The result does not give the answer to the question whether it was the peptide or just electroporation that reduced the whole cell lysate protein content. Additionally, the experiment with scrambled (or random or mutant) peptide/peptides is required.

The reviewer is correct. In this revised manuscript, we have removed these data as per the suggestion of reviewer 2.

     14.Figure 3: The design of the experiment seems to be inadequate and lacks the reasoning. Why should the short peptide have the function of the whole FAM83G protein? Please explain e.g. by providing reasoning based on protein domains/interactions etc.

The reviewer is correct. In order to support our contention that the short FAM83G peptide induces apoptosis due to reduction of HSP27 S15 and S82 phosphorylation, we have now compared the wild type peptide with a mutant peptide in which S356 is changed to alanine (S356A-AF-956) that is resistant to phosphorylation by PKD1/PKCμ. Those new data are now represented in new Figure 4.

      15.Figure 4a: What was the vehicle control?

Rather than vehicle control we used a random peptide (same amino acid composition but in a random order) used in the same vehicle that we termed AF-859. In the revised manuscript, the Figure number is now changed to Figure 3a.

     16.Figure 4b: The untreated control is missing. Without that, one cannot say “treatment with 54 μM AF-859 (closed column) or 54 μM AG-066 (vertical 25 striped column) had no significant effect on the cell number.”

We confirmed that there was no difference between AF-859 and vehicle treated condition in terms of living cell in preliminary experiments.

      17.Figure 4c: Immunoblot seems to be overexposed, lacks the untreated sample, and lacks loading control.

As described above as response #15, we confirmed that there was no difference between AF-859 and non-treated condition in terms of living cell by preliminary experiment. The loading control is shown as a-tubulin in Figure 3f (In the revised manuscript, new Figure number is 3 instead of 4) and we switched the blotting to the lighter exposure one as shown in Figure 3C.

     18.Figure 4d: Please explain the reason why the overexpression of FLAG-FAM83G protein does not change the level of HSP27 expression (Figure 1b), and the treatment with AF- 956 peptide gives such a HSP27 protein level increase.

The reviewer has brought up an important point. When FAM83G was transiently overexpressed by electroporation, the increase in FAM83G overexpression started at 2 hours and reached to maximum levels by 48 hours. In the case of peptide experiment, we could not determine the amount of peptide that entered into cells was equal to that overexpressing FAM83G by electroporation. We were also not able to determine when the introduced peptide reached the maximum levels. We speculated that cells compensated by increasing HSP27 protein levels to prevent cell death.

    19.Figure 4d: Loading control is missing.

Since the same gel membrane transfers were cut for different molecular weights and were also stripped and re-probed, the a-tubulin is the loading control for these immunoblots.

   20.Figure 4d: Untreated control is missing.

The reviewer is correct. In fact, we did preliminary experiments to compare AF-859 treated cells and non-treated cells and confirmed that there are no differences. Also, we carefully titrated the PFA concentration to avoid toxicity during preliminary experiments.

    21.Figure 4e, 4f: The experiments require additional control with scrambled (or random or mutant) peptide/peptides containing similar colon cancer entry signal as A-956 to prove for the pro-apoptotic activity of this particular peptide.

We synthesized S356A-AF-956 peptide for a complemental study. As shown in a new Figure 4, S356A-AF-956 peptide did not show apoptosis phenomenon.

    22.Figure 5a: The untreated control is missing.

Please see response to criticism 19.

    23.Figure 5b: Immunoblot lacks the untreated sample, and lacks loading control.

Please see response to criticism 18.

    24.Figure 5c: Untreated control is missing.

Please see response to criticism 19.

    25.Figure 5d: The experiments require additional control with scrambled (or random or mutant) peptide/peptides containing similar liver cancer entry signal as AG-066 to prove for the pro-apoptotic activity of this particular peptide.

Please see response to criticism 19.

    26.Line 294-296: The statement ‘FAM83G protein level in cancerous cells (HCT116 and HepG2 cells) was apparently higher than that in noncancerous cells (CHO cells) cannot be acknowledged. HCT116 and HepG2 cells are human cells, while CHO are hamster cells. One cannot compare relative expression levels of the proteins from different species using WB technique without proving, that in all the used species the antibody binds to the targeted proteins with similar affinities.

The reviewer is absolutely correct. We have addressed this point in line 6 on page 18-line 3 on page 19.

     27.Figure 7b, lines 338-339: ‘Thus, it was confirmed that FAM83G-rich cancer cells possess a large amount of HSP27’ – this does not seem true. The figure shows that when FAM83G mRNA level is ‘smaller’, HSP27 mRNA is high, and when FAM83G mRNA is ‘larger’, HSP27 mRNA is lower. Therefore, there is rather a negative correlation, if there is any.

We are sorry we miss-labeled the original bar graph. This has now been corrected and presented as a new Figure 8b.

Reviewer 2 Report

In this manuscript molecules-731148, authors Okada et al. have explored the possible role of the protein FAM83G as an apoptosis inducer, using studies in various cell lines.

Overall, the study is sound and the manuscript is well written with a clear presentation of data. However some issues need to be addressed, as noted below:

1) Results section: where the authors give the consensus peptide sequence for Ser-phosphorylation on FAM83G, it would be helpful to highlight which serine corresponds to S356 (since there are two Ser residues in the peptide).

2) The sentence “therefore firstly the FAM83G protein was found to co-ip with PKD” sounds inappropriate. Correct usage would be “therefore we wanted to test whether FAM83G was able to co-ip with PKD”.

3) Figure 1b: lanes need to be labeled, in addition to the description for each lane provided in the legend, otherwise it makes it hard to follow.

4) In addition to using the S356A resistant mutant form of FAM83G, the authors should use an S82A phospho-resistant form of Hsp27. If their hypothesis is indeed correct, then the expression levels of WT-FAM83G should have no effect on the levels of Hsp27 (as seen in their Figure 1b). I think this will be an important control to prove their hypothesis.

5) Figure 2: since the p-value is 0.067, how do the authors say that the cell number in HCT116 cells was “significantly reduced”?

6) Figure 4: all the bar graphs need to be bigger, and at better resolution.

7) In their studies with synthetic peptides, I am not convinced that using the AF-859 version was necessary. Also, I think it is absolutely necessary to include a negative ‘control’ peptide similar to AF-956, but with Ser mutated to Ala (similar to their S356A FAM83G mutant). That would be a more suitable negative control.

8) Figure 7b: With the error bar as big as shown, I am not convinced that the p-value can be as low as 0.0066. The authors need to check their statistical analysis.

I think all these issues need to be addressed, before this manuscript can be accepted for publication to this journal.

Author Response

Reviewer 2

Comments and Suggestions for Authors

In this manuscript molecules-731148, authors Okada et al. have explored the possible role of the protein FAM83G as an apoptosis inducer, using studies in various cell lines. Overall, the study is sound and the manuscript is well written with a clear presentation of data. However, some issues need to be addressed, as noted below:

1) Results section: where the authors give the consensus peptide sequence for Ser phosphorylation on FAM83G, it would be helpful to highlight which serine corresponds to S356 (since there are two Ser residues in the peptide).

As suggested, we have added an underline below S356 and S356A.

2) The sentence “therefore firstly the FAM83G protein was found to co-ip with PKD” sounds inappropriate. Correct usage would be “therefore we wanted to test whether FAM83G was able to co-ip with PKD”.

We rewrote these part as “endogenous FAM83G protein was co-immunoprecipitated with endogenous PKD1/PKCμ (pCMV6 control lane). Overexpressed FAM83G co-immunoprecipitated with increased amount of PKD1/PKCμ compared to pCMV6 samples as shown in pCMV6-wild type (WT)-FAM83G lane. These data suggested that FAM83G is associated with PKD1/PKCμ.” in line 1-6 on page 9.

3) Figure 1b: lanes need to be labeled, in addition to the description for each lane provided in the legend, otherwise it makes it hard to follow.

As requested, we have now added labels to the lanes as x10 times, x5 times, and x0 times as comparison with endogenous FAM83G in Figure 1b.

4) In addition to using the S356A resistant mutant form of FAM83G, the authors should use an S82A phospho-resistant form of Hsp27. If their hypothesis is indeed correct, then the expression levels of WT-FAM83G should have no effect on the levels of Hsp27 (as seen in their Figure 1b). I think this will be an important control to prove their hypothesis.

Unfortunately, we do not have the S82A-HSP27 construct. As an alternative approach, we performed S356A-AF-956 and S356A-AG-066 peptides experiments. As shown in new Figure 4 and 6, S356A-AF-956 and S356A-AG-066 neither induce apoptosis nor S15 and S82 HSP27 phosphorylation. Also, S356A-AF-956 and S356A-AG-066 did not show caspase 3 and PARP protein cleavage. Taken together, these data further support our conclusion that FAM83G functions as a spontaneous apoptosis inducer by decreasing the phosphorylation of HSP27 at S15 and S82.

5) Figure 2: since the p-value is 0.067, how do the authors say that the cell number in HCT116 cells was “significantly reduced”?

This was a typographical error. The correct p value is 0.0006.

6) Figure 4: all the bar graphs need to be bigger, and at better resolution.

We are sorry about that. We tried to show bar graphs as bigger size as much as we can.

7) In their studies with synthetic peptides, I am not convinced that using the AF-859 version was necessary. Also, I think it is absolutely necessary to include a negative ‘control’ peptide similar to AF-956, but with Ser mutated to Ala (similar to their S356A FAM83G mutant). That would be a more suitable negative control.

As an alterative to the S356A-FAM3G mutant, we performed new experiments using S356A-AF-956 and S356A-AG-066 peptides. These data are shown in new Figure 4 and 6 respectively.

8) Figure 7b: With the error bar as big as shown, I am not convinced that the p-value can be as low as 0.0066. The authors need to check their statistical analysis.

The reviewer is correct and we apologize for this error. The correct p value is “<0.05”.

I think all these issues need to be addressed, before this manuscript can be accepted for publication to this journal.

We hope that we have sufficiently addressed all of your concerns now and that the revised manuscript will now be considered acceptable for publication in the journal.

Round 2

Reviewer 2 Report

In this revised draft, the authors Okada et al have satisfactorily addressed all my concerns from before.

I believe that the manuscript is now acceptable for publication in this journal.